

# A possible new spawning area for Atlantic bluefin tuna (*Thunnus thynnus*): the first histologic evidence of reproductive activity in the southern Gulf of Mexico

Roberto Cruz-Castán[1,2], Sámar Saber[3], David Macías[3], María José Gómez Vives[3], Gabriela Galindo-Cortes[2], Sergio Curiel-Ramirez[4] and César Meiners-Mandujano[2]

[1] Posgrado en Ecología y Pesquerías, Universidad Veracruzana, Boca del Río, Veracruz, Mexico
[2] Instituto de Ciencias Marinas y Pesquerías, Universidad Veracruzana, Boca del Río, Veracruz, Mexico
[3] Centro Oceanográfico de Málaga, Instituto Español de Oceanografía, Fuengirola, Malaga, Spain
[4] Instituto de Investigaciones Oceanológicas, Universidad Autónoma de Baja California, Ensenada, Baja California, Mexico

Corresponding authors
Roberto Cruz-Castán,
cas213@hotmail.com
César Meiners-Mandujano,
cmeiners@uv.mx

## ABSTRACT

The number of studies of reproductive biology for Atlantic bluefin tuna carried out in the Gulf of Mexico is significantly lower than those undertaken in the Mediterranean Sea. Four spawning areas have been found for the eastern Atlantic bluefin tuna stock in the Mediterranean Sea, so it is not implausible that there is more than one spawning area in the Gulf of Mexico for the western Atlantic bluefin tuna stock. The individuals used in this study were caught as bycatch by the Mexican surface longline fleet between January and April 2015. A total of 63 individuals ranging between 192 and 293 cm $L_F$ (mean = 238 ± 22.52 cm) were measured. Gonads from 46 fish (31 females and 15 males) were collected for histological examination. All the individuals were classified as mature; 25 were reproductively active (in spawning capable and spawning stages). The histological analysis indicates spawning activity in Mexican waters (the southern Gulf of Mexico). Spawning occurred in March and April, when the sea surface temperature was 25.57 °C ± 0.69 in March and 27.03 °C ± 0.69 in April. Information on the location of the spawning areas is necessary for a correct management of species. The present study provides the first histological evidence of reproductive activity in Mexican waters, and indicates a wider spawning area, beyond just the northern zone, potentially encompassing the entire Gulf of Mexico.

## INTRODUCTION

Atlantic bluefin tuna *Thunnus thynnus* (*Linnaeus, 1758*) is a large, highly migratory species distributed in the Atlantic Ocean between 70° N and 30° S latitudes (*Collette & Nauen, 1983*). The International Commission for the Conservation of Atlantic Tunas (ICCAT) categorizes two different bluefin tuna stocks for management purposes, the eastern and the western Atlantic stocks, separated at the 45° W meridian. This borderline was based on the

two well-known spawning areas, the Mediterranean Sea and Gulf of Mexico. Atlantic bluefin tuna is classified as an endangered species by the International Union for Conservation of Nature as a consequence of fishing pressure (*Collette et al., 2011a*, *2011b*). Currently, there are ICCAT regulations aimed at managing both stocks (https://www.iccat.int/en/RecRes.asp). Although the western stock was the first to be under regulation (since 1999), the number of reproductive studies is significantly lower than those undertaken for the eastern stock (*Susca et al., 2001*; *Corriero et al., 2003*; *Aranda et al., 2011*; *MacKenzie & Mariani, 2012*).

The studies of the western stock focus on Canadian waters and the northern region of the Gulf of Mexico, specifically in United States waters (*Heinisch et al., 2014*; *Knapp et al., 2014*). The spawning area was located in the north and northwest of the Gulf of Mexico (*Nemerson, Berkeley & Safina, 2000*; *Ingram et al., 2010*). Over the years it has been thought that the western Atlantic bluefin tuna spawned only in the north Gulf of Mexico, but a recent larval study has reported a new spawning ground in the Slope Sea (*Richardson et al., 2016*). Furthermore, the southern Gulf of Mexico, that is, the Mexican waters according with *Ward & Tunnell (2017)*, has been suggested to be a potential spawning area for Atlantic bluefin tuna due to the marked seasonality of the large individuals caught which arrive to the southern of the Gulf of Mexico when the sea surface temperature (SST) registers the optimum thermal window to carry out reproductive activity (*Abad-Uribarren et al., 2014*).

Information on the location of the spawning areas is necessary for a correct management of species. In the Mediterranean Sea four spawning zones have been described for the eastern stock (*Karakulak et al., 2004*), so it is quite plausible that more than one spawning area in the Gulf of Mexico can exist. Until now there have been no studies on the reproductive status of individuals at the time of their appearance in Mexican waters. The main objective of this work was, through the analysis of size frequency, and histological examination of the gonads, to determine the maturity status of western Atlantic bluefin tuna caught in Mexican waters.

## MATERIALS AND METHODS

The individuals used in the present work were caught as bycatch by the Mexican surface longline fleet targeting yellowfin tuna (*Thunnus albacares*) from January to April 2015. Both the scientific observers on board and the surveys of the fishermen conducted throughout 2015 reported that no Atlantic bluefin tuna were caught between May and December. Atlantic bluefin tuna individuals were measured to the nearest cm (fork length; $L_F$) by scientific observers on longline vessels. Length analysis was conducted using kernel density estimators (KDE) (*Salgado-Ugarte, 2002*; *Rivera-Velázquez et al., 2010*). The KDE equation is

$$\hat{f}(x) = \frac{1}{nh} \sum_{i=1}^{n} K\left(\frac{x - X_i}{h}\right)$$

where $\hat{f}(x)$ is the density estimation of the variable $x$, $n$ is the number of observations, $h$ is the bandwidth, $X_i$ corresponds to length of the $i$-th fish specimen and $K$ is a smooth, symmetric kernel function integrating to one.

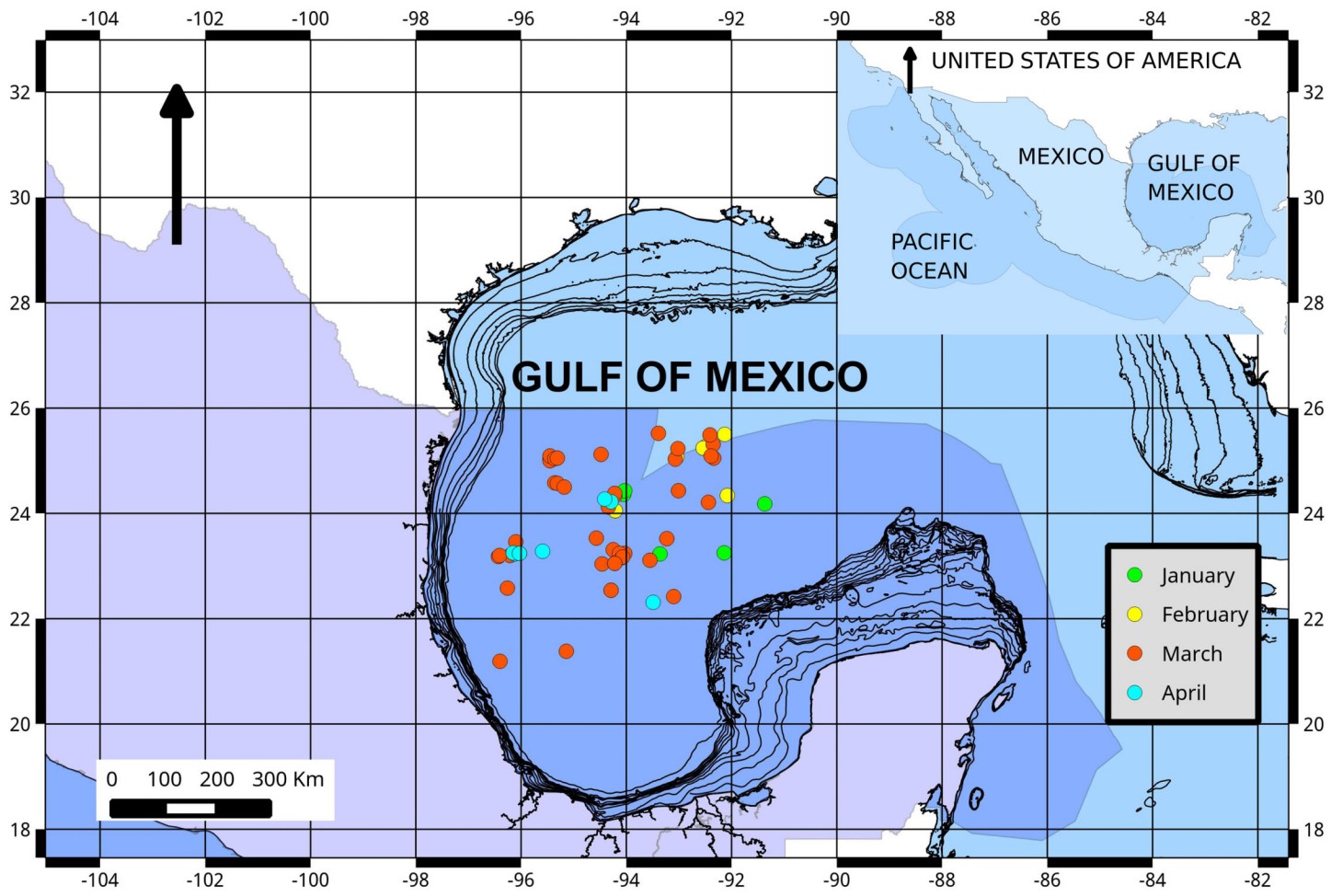

**Figure 1** **Fishing ground locations of Atlantic bluefin tuna caught by month (2015) in the southern Gulf of Mexico.** The dark blue shading represents the Exclusive Economic Zone (EEZ) of Mexico.

The location of each individual was recorded (Fig. 1) and a subset of Atlantic bluefin tuna was sampled to collect gonad tissues. Gonad samples were fixed in Bouin's liquid for 4 h and subsequently preserved in 70% ethanol. A preserved gonad subsample was embedded in paraffin, sectioned at 10 μm and stained with Mallory's trichrome stain. Five or three histological sections separated 400 μm apart were examined per female and male, respectively. Microscopic classification used for Atlantic bluefin tuna gonads was based on a modification of the criteria of *Schaefer (1998)* and *Farley et al. (2013)*.

Six developmental oocyte stages were considered in this study: primary growth, lipid-stage, early vitellogenic, advanced vitellogenic, migratory nucleus (MG), and hydrated (HY) oocytes. The most advanced group of oocytes (MAGO) present within each ovary and, the presence/absence of: postovulatory follicles (POFs), atretic follicles, and late stages of atresia were used to determine the sexual maturity. Females were considered as mature if ovaries contained vitellogenic, MG or HY oocytes and/or atresia, and were classified into seven ovary stages (Table 1).

**Table 1 Histological classification of gonad stages for female Atlantic bluefin tuna (*Thunnus thynnus*).**

| Maturity status | Activity | Gonad stage | MAGO | Additional features |
|---|---|---|---|---|
| Immature | Inactive | Immature | PG or LS | • Absence of atresia<br>• No POFs |
| Mature | Inactive | Developing | E-Vit | • Some atresia of vitellogenic oocytes present<br>• No POFs |
| Mature | Active | Spawning capable | A-Vit | • Some atresia of vitellogenic oocytes may be present<br>• No POFs |
| Mature | Active | Spawning | A-Vit | • POFs are present<br>• Some atresia may be present |
| Mature | Active | Spawning | MG or HY | • POFs could be present<br>• Some atresia may be present |
| Mature | Inactive | Regressing | LS, E-Vit, or A-Vit | • Abundant atretic follicles<br>• No POFs<br>• Disorganization of ovary structures, with some spaces |
| Mature | Inactive | Regenerating | PG or LS | • Late stages of atresia<br>• No POFs |

**Note:**
MAGO, The most advanced group of oocytes; oocyte stages: PG, primary growth; LS, lipid-stage; E-Vit, early vitellogenic; A-Vit, advanced vitellogenic; MG, migratory nucleus; and HY, hydrated; oocytes; POFs, postovulatory follicles.

For males, four cellular stages, namely spermatogonia, spermatocytes, spermatids, and spermatozoa (SZ), were microscopically differentiated and recorded. Five testes stages were then assigned based on: the relative abundance of cysts containing the four cellular stages, the presence or absence of SZ within seminiferous tubules, and the amount of sperm (when present) within the central longitudinal sperm duct (vas deferens) (Table 2).

Monthly data of the SST were obtained from the telematics interface for the visualization and analysis of data of "Giovanni" (*Acker & Leptoukh, 2007*) remote detection from a satellite with a spatial resolution of four km. A temporal series of the monthly average of the SST was built from a regular polygon that included the area of the captures to correlate the mean SST with the different reproductive stages.

## RESULTS

Although Mexican longline fishery targeting yellowfin tuna operates all over the year, bluefin tuna catches only occurred between January and April. A total of 63 individuals were caught, five in January (7.93%), eight in February (12.69%), 42 in March (66.66%), and eight in April (12.69%). Sizes of these individuals ranged from 192 to 293 cm $L_F$, with a mean of 238 ± 22.52 cm $L_F$. The size structure was determined by a dominant mode of 235 cm $L_F$ (Fig. 2). Gonads of 46 specimens, 31 ovaries, and 15 testes, were histologically

**Table 2 Histological classification of gonad stages for male Atlantic bluefin tuna (*Thunnus thynnus*).**

| Gonad stage | Main features |
|---|---|
| Immature or virgin | • Only SG present<br>• No sperm in the sperm duct<br>• Small space of lobule lumen |
| Early spermatogenesis or developing | • Cysts with SG<br>• Some SC and SD<br>• Some SZ |
| Late spermatogenesis or spawning capable | • Abundant SD<br>• Some SZ within seminiferous tubules |
| Spawning | • Some SD<br>• Plenty of SZ<br>• Sperm duct full of sperm |
| Spent or regressing | • Residual SZ |

**Note:**
Spermatogonia (SG), spermatocytes (SC), spermatids (SD), and spermatozoa (SZ).

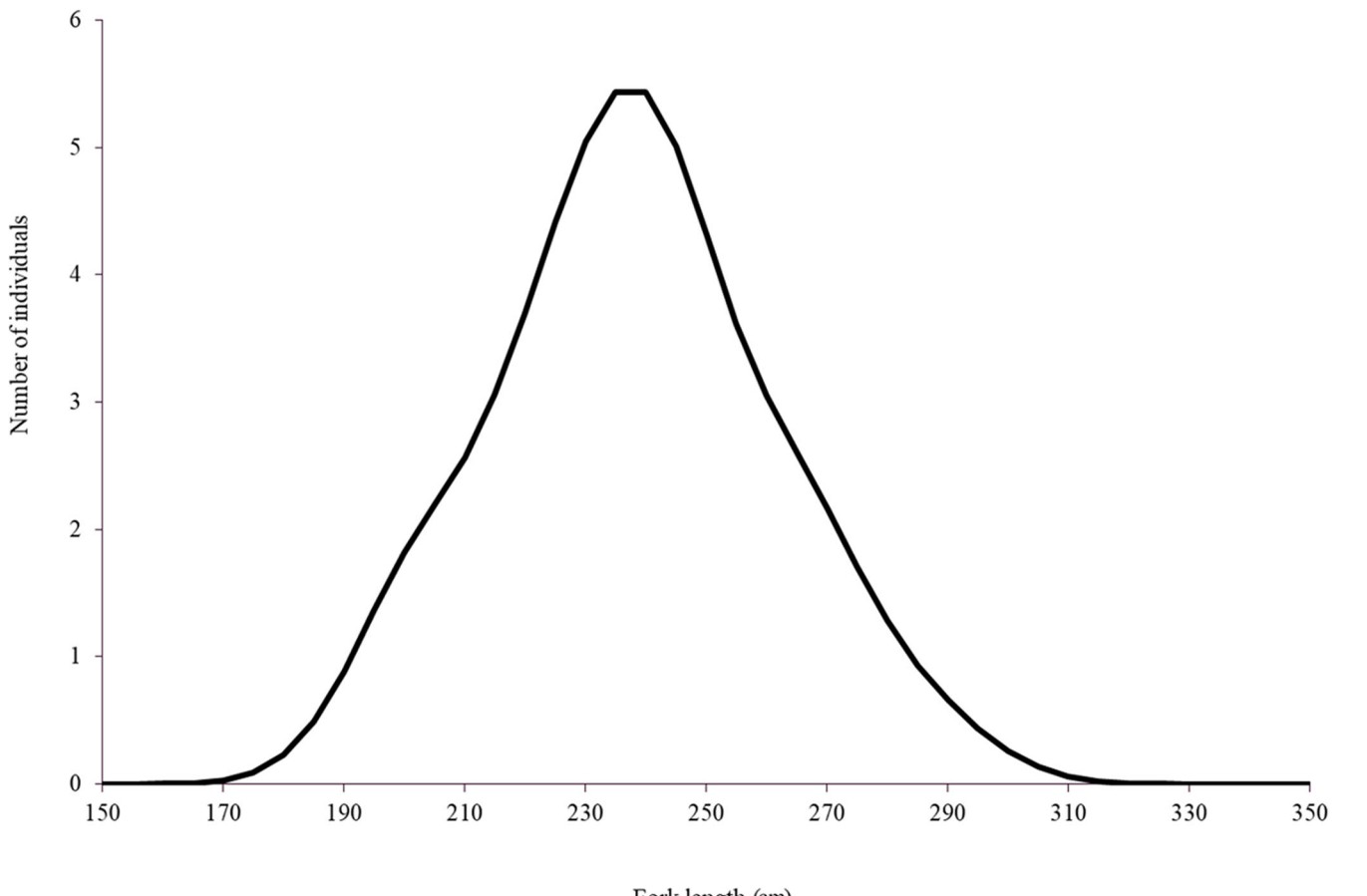

**Figure 2 Length distribution of Atlantic bluefin tuna caught by the Mexican surface longline fleet targeting yellowfin tuna on 2015 (*n* = 63).**

**Table 3 Temporary progression of gonad stages for mature individuals (females and males) of Atlantic bluefin tuna (*Thunnus thynnus*) collected in 2015.**

| Reproductive stage | Months | | | | | | | | | | | | Total |
|---|---|---|---|---|---|---|---|---|---|---|---|---|---|
| | January | | | February | | | March | | | April | | | |
| | Early | Mid | Late | Early | Mid | Late | Early | Mid | Late | Early | Mid | Late | |
| Female | | | | | | | | | | | | | |
| Regenerating | 1 | 1 | | | | 4 | 3 | 1 | 2 | | | | 12 |
| Developing | | | | | | | | | 1 | 1 | | | 2 |
| Spawning capable | | | | | | | | | 12 | 2 | | | 14 |
| Spawning | | | | | | | | | | | 1 | | 1 |
| Regressing | | | | | | | | | 2 | | | | 2 |
| Male | | | | | | | | | | | | | |
| Early spermatogenesis | | 1 | | | | | 1 | | | | | | 2 |
| Late spermatogenesis | | | | | | | | | 5 | 1 | | | 6 |
| Spawning | | | | | | | | 2 | 1 | | | 1 | 4 |
| Spent | | | | | | | | | 1 | 2 | | | 3 |

Note:
Months were divided in early (from the day 1 to 10), mid (from the day 11 to 20), and late (from the day 21 to 31).

examined to determine their temporary progression of reproductive stages (Table 3). Only mature individuals were found in the present study. The five ovarian stages observed are shown in Fig. 3. All ovaries collected in January and February (2 and 4, respectively) were in regenerating stage. In March, six ovaries in regenerating stage were also observed (29%), one ovary (5%) was in developing, whereas those in the spawning capable stage were found to be more frequent with 12 ovaries (57%) and 2 (10%) were in regressing stage. In April, one ovary (25%) was in developing, two ovaries (50%) were in spawning capable and one (25%) was in spawning. No POFs were observed, the ovary classified as spawning showed MG oocytes as MAGO. The four testes stages found are shown in Fig. 4. Only one male was collected in January, being in early spermatogenesis. No testes were collected in February. In March, one testis (10%) was in early spermatogenesis, five testes (50%) were in late spermatogenesis, three (30%) were in spawning, and one testis (10%) was in regressing (spent). In April, one testis (25%) was in late spermatogenesis, one (25%) was in spawning, and two testes (50%) were in regressing (spent).

Sea surface temperature in the southern Gulf of Mexico was increasing slightly from January to February, from 23.04 °C ± 0.69 to 23.42 °C ± 0.69 in March a temperature of 25.57 °C ± 0.69 was registered and finally in April the SST reached 27.03 °C ± 0.69 (mean ± SE; Fig. 5).

## DISCUSSION

This is the first study that reports histological information for reproductive status of Atlantic bluefin tuna in the southern Gulf of Mexico. According to *Abad-Uribarren et al. (2014)* our results suggest a seasonality of the bluefin tuna bycatch in the Mexican longline fishery targeting yellowfin tuna. The timing of the catches for this species (7.9% of catches in January, 12.7% in February, increasing substantially to 66.7% in March, and

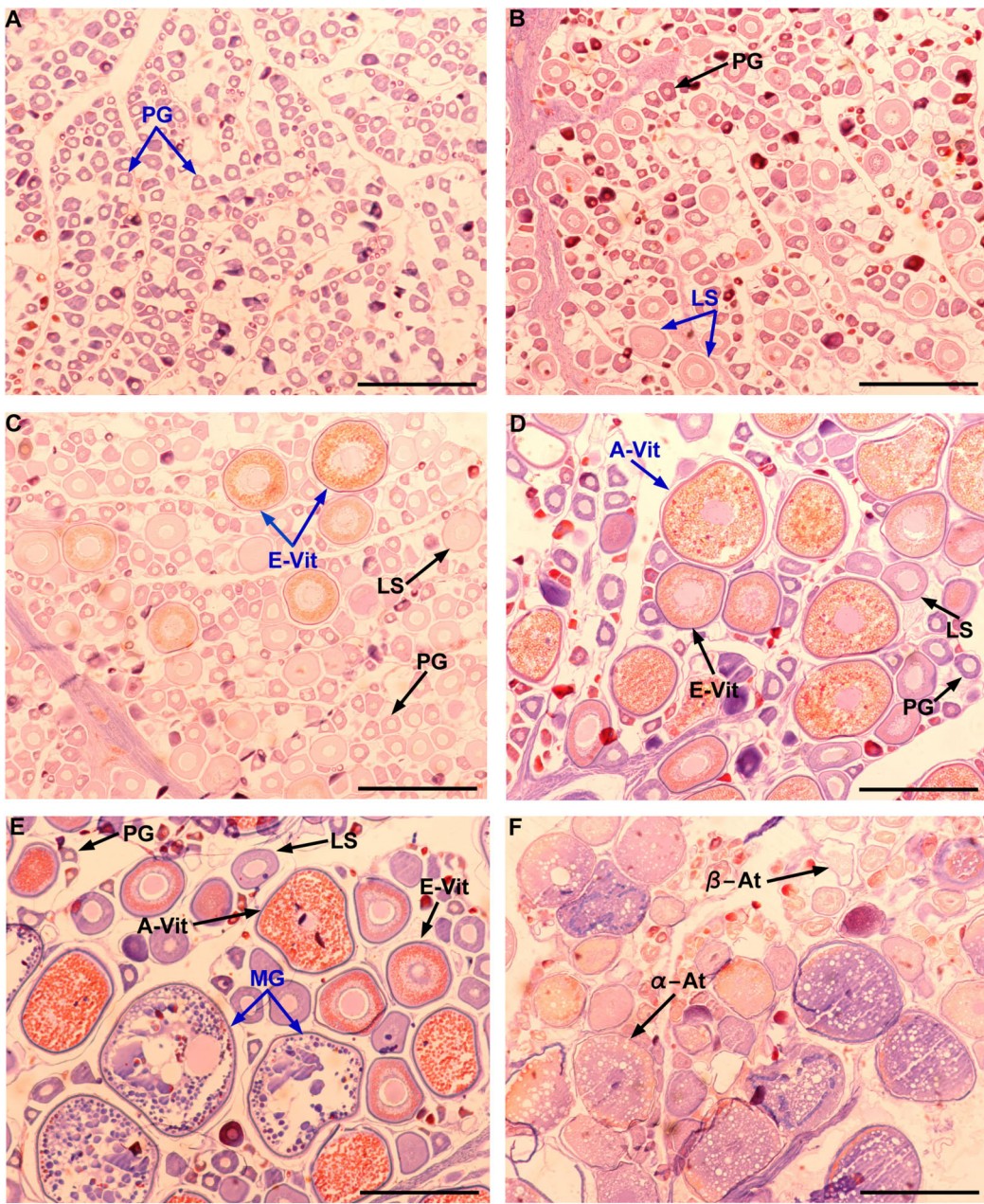

**Figure 3 Gonad stages for female individuals of Atlantic bluefin tuna caught in the southern Gulf of Mexico.** Ovarian in regenerating (A–B), developing (C), spawning capable (D), spawning (E), and regressing stages (F). PG, primary growth oocyte; LS, lipid-stage oocyte; E-Vit, early vitellogenic oocyte; A-Vit, advanced vitellogenic oocyte; MG, migratory nucleus oocyte; α-At, alpha atresia; β-At, beta atresia. The MAGO for each stage are indicated in blue color. Scale bar = 500 μm.

decreasing in April to 12.7%) suggests that Atlantic bluefin tuna gradually arrive to Mexican waters in January and February, registering the highest catch in March perhaps due to increased feeding behavior before the spawning season and a decrease in April when spawning begins. Although there are no previous studies regarding the feeding patterns

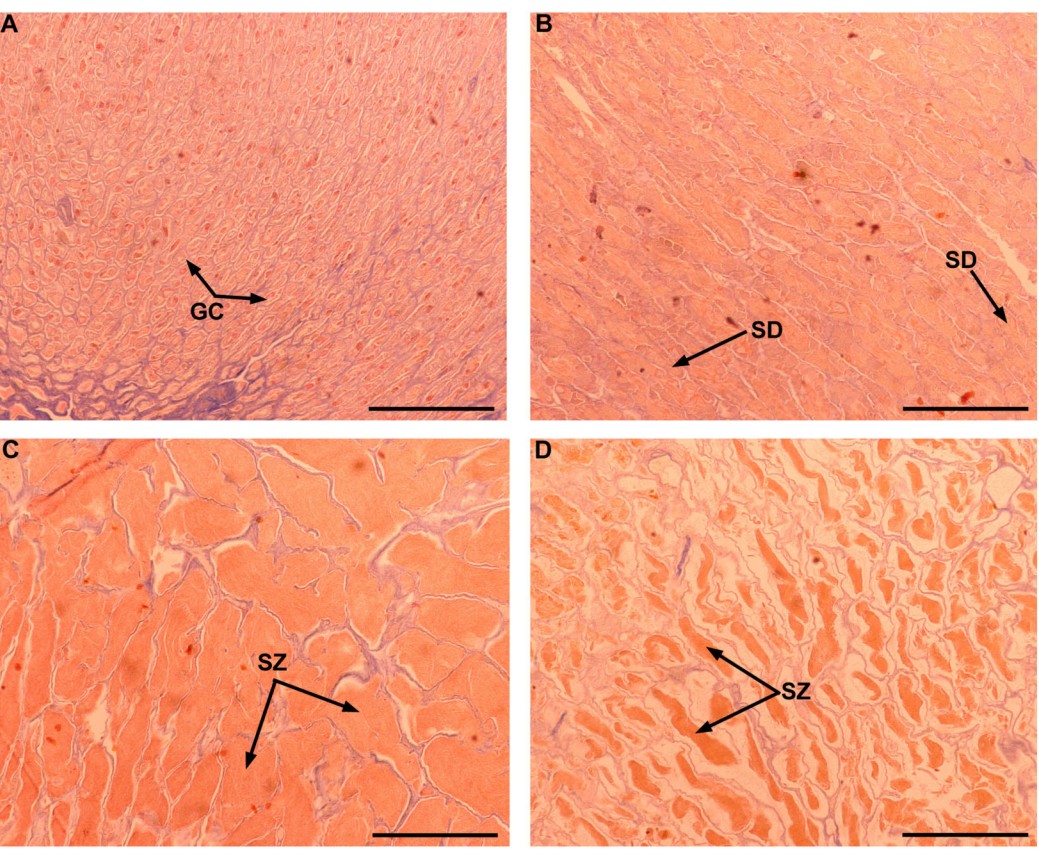

**Figure 4 Gonad stages for male individuals of Atlantic bluefin tuna caught in the southern Gulf of Mexico.** Testes in early spermatogenesis (A), late spermatogenesis (B), spawning (C), and spent stages (D). GC, germinal cells; SD, spermatids; SZ, spermatozoa. Scale bar = 500 µm.

for this species at the southern Gulf of Mexico, a similar behavior has been previously described for Pacific bluefin tuna *Thunnus orientalis* by *Chen, Crone & Hsu (2006)* who found a decrease in feeding when the spawning period starts, as well as for other tuna species (*Rivas, 1954*).

It is known that the reproductive season of tunas is strongly linked with temperature and 24 °C is ideal for spawning (*Schaefer, 1998*). The SST registered in the fishery zone is 25.57 °C ± 0.69 in March and 27.03 °C ± 0.69 in April, in agreement with SST in March and April reported in the northern spawning zone of Atlantic bluefin tuna in the Gulf of Mexico, where larvae were found from 25 to 28 °C (*Muhling, Lamkin & Roffer, 2010*).

Several studies indicate that the spawning period of Atlantic bluefin tuna is about 3 months, from May to July for eastern stock and from April to June for western stock (*Clay, 1991*; *Knapp et al., 2014*). According to *Diaz & Turner (2007)* the sizes of the individuals caught in this study from January to April correspond to sexually mature individuals. The histological examination of gonads showed that 48% of female and 67% of males were reproductively active. Male individuals in spawning stage were found after mid-March and males in regressing stage with evidence of residual SZ from previous spawning were

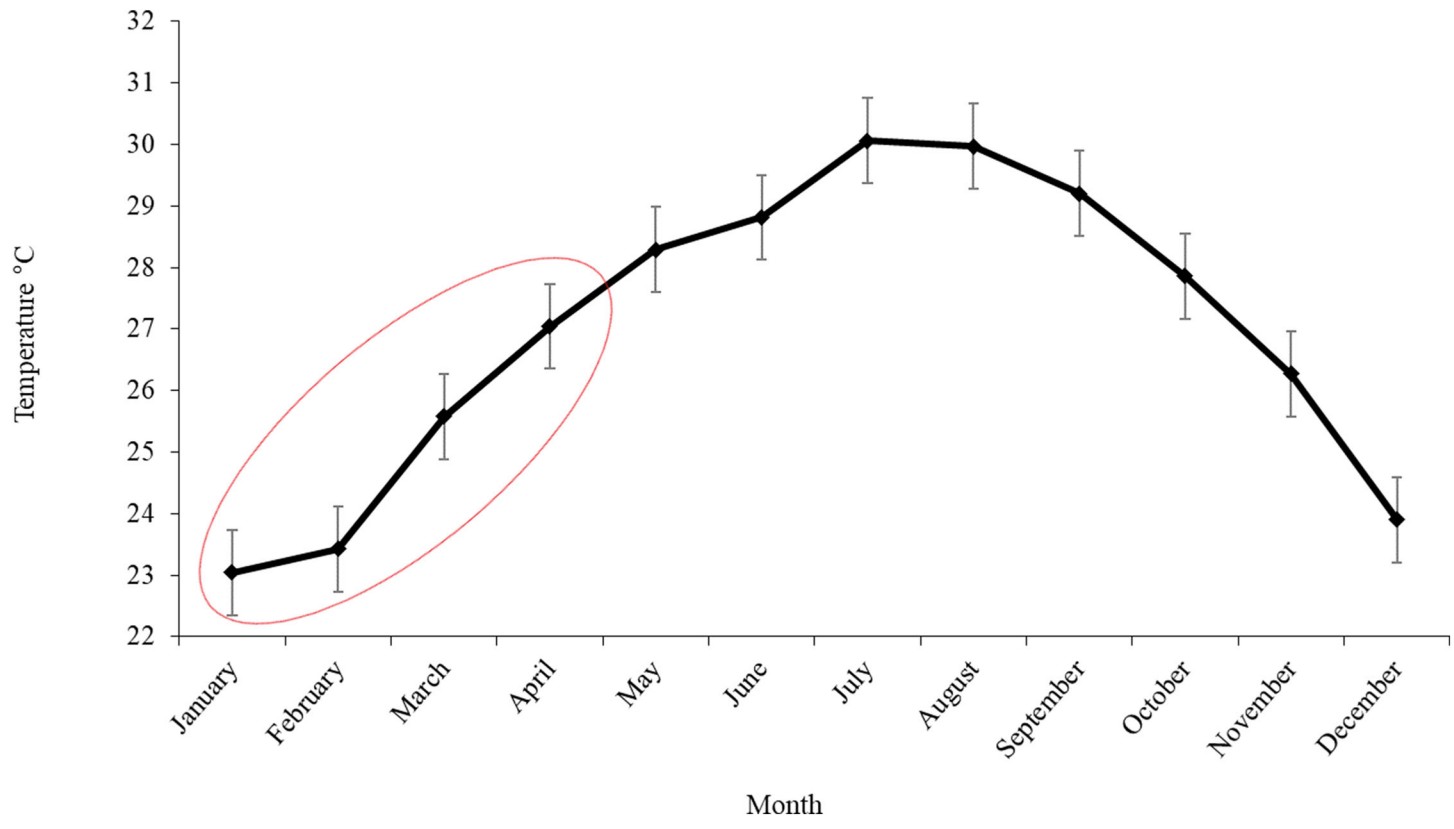

**Figure 5 Monthly variation of SST in the area of operation of the Mexican longline fleet targeting yellowfin tuna during 2015.** The ellipse shows the temperature in the months at which Atlantic bluefin tuna were caught. The error bars represent the standard error (SE).

found in late March and early April. For females, the most frequent ovary stage in March was spawning capable and one female in spawning stage was found in April, corresponding with the spawning season for western stock (*Schaefer, 2001*; *Teo, Boustany & Block, 2007*). *Lutcavage et al. (1999)* expressed the need to consider other possible spawning areas for western stock and discuss the possibility of a spawning zone in the mid-Atlantic region, with similar hydrographic characteristics to the spawning area of the north Gulf of Mexico. However, they could not prove the existence of a new spawning zone due to the lack of histological evidence. In the present study individuals in spawning capable ($n = 20$) and in spawning ($n = 5$) stages have been found in Mexican waters, suggesting a wider spawning area, beyond just the northern zone, potentially encompassing the entire Gulf of Mexico.

## CONCLUSIONS

Additional studies are needed to support the southern Gulf of Mexico as a habitual spawning area. However four pieces of evidence to show that the southern Gulf of Mexico could be part of the spawning zone for Atlantic bluefin tuna and therefore suggest the possibility that the entire Gulf of Mexico may be a spawning zone for this species: (1) There is a marked seasonal occurrence of individuals, (2) SST is appropriate to carry out the

reproduction of this species, (3) the sizes of fish caught correspond to sexually mature individuals, and (4) the histological analysis of gonads shows individuals in active reproductive stages (spawning capable and spawning).

## ACKNOWLEDGEMENTS

The authors would like to recognize the effort and support of K&B tuna company, in particular to the CEO José Bisteni for all his support, all fishermen and staff from longline fleet of Tuxpan port, Veracruz, Mexico. The authors also thank Tim Dobinson for the English language proof reading of this manuscript and M.C. Ana Gabriela Galicia Cruz for her support in editing figures. We would like to thank the two anonymous reviewers for the help with the revision. Analyses and visualizations of SST used in this paper were produced with the Giovanni online data system, developed and maintained by the NASA GES DISC.

### Funding

Roberto Cruz-Castán received financial support from a CONACYT postgraduate scholarship No. 408210/257994. There was no additional external funding received for this study. The funders had no role in study design, data collection and analysis, decision to publish, or preparation of the manuscript.

### Grant Disclosures

The following grant information was disclosed by the authors:
CONACYT postgraduate scholarship: 408210/257994.

### Competing Interests

The authors declare that they have no competing interests.

### Author Contributions

- Roberto Cruz-Castán conceived and designed the experiments, performed the experiments, analyzed the data, contributed reagents/materials/analysis tools, prepared figures and/or tables, authored or reviewed drafts of the paper, approved the final draft.
- Sámar Saber conceived and designed the experiments, performed the experiments, analyzed the data, contributed reagents/materials/analysis tools, prepared figures and/or tables, authored or reviewed drafts of the paper, approved the final draft, histological classification.
- David Macías conceived and designed the experiments, performed the experiments, analyzed the data, contributed reagents/materials/analysis tools, prepared figures and/or tables, authored or reviewed drafts of the paper, approved the final draft, histological classification.
- María José Gómez Vives conceived and designed the experiments, performed the experiments, analyzed the data, contributed reagents/materials/analysis tools, authored or reviewed drafts of the paper, approved the final draft.
- Gabriela Galindo-Cortes analyzed the data, contributed reagents/materials/analysis tools, prepared figures and/or tables, authored or reviewed drafts of the paper, approved the final draft, kernel density estimators (EDK).
- Sergio Curiel-Ramirez analyzed the data, contributed reagents/materials/analysis tools, authored or reviewed drafts of the paper, approved the final draft.
- César Meiners-Mandujano analyzed the data, contributed reagents/materials/analysis tools, prepared figures and/or tables, authored or reviewed drafts of the paper, approved the final draft.

## Data Availability

The raw data used in this study are available in File S1. These raw data show information of temperature and EDK values for the length analysis.

## Supplemental Information

Supplemental information for this article can be found online at http://dx.doi.org/10.7717/peerj.7187#supplemental-information.

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
