# Peer review of "A possible new spawning area for Atlantic bluefin tuna (Thunnus thynnus): the first histologic evidence of reproductive activity in the southern Gulf of Mexico"

_PeerJ, doi:10.7717/peerj.7187_

## Round 0.1 · original submission · Major Revisions

The authors need to respond to the comments provided by reviewers, providing more detail about the histological interpretation, and be more cautious in their interpretation of the data, given the limited dataset.

Some additional queries/comments are:

What is the expected sex ratio in tuna populations? Is having twice as many females as males significant?
Fig 1 – should provide an indication of where the northern GoM is, as compared to where fish in this study were caught (southern GoM)
Fig 2 – the curve is significantly smoothed so the legend should indicate the total number of fish that were used in the plot.
Fig 3 – are these images all from different individuals? How many sections were examined per fish? The methods should elaborate, and advanced group of oocytes (MAGO) should be cross-referenced in the figure.
Fig 5 – what is the source of SST data?
Table – rather than put early, mid, late, why not list the exact days and give readers this explicit information?
Table 1 and 2 could be merged.

Reviewer 1 ·

Basic reporting

There are a number of issues with the reporting of the results of the study:
(1) title: the title needs to refer to Atlantic bluefin tuna
(2) abstract: the first sentence needs to be refer to Atlantic bluefin tuna. Also, the abstract does not explain why it is important to know whether spawning occurs in Mexican waters (e.g. the importance of this information for fisheries management); the abstract needs a concluding sentence;
(2) line 42: explain the meaning of ICCAT;
(3) lines 51-52: there needs to be some more biological background to the sentence (i.e. it suggests that the tuna migrate into Mexican waters but no information on migration pathways is provided in the Introduction, therefore the relevance of knowing this information is not obvious);
(4) the Introduction does not conclude with statements about why the information collected by this study is needed or important;
(5) some of the information in some figures is unclear: Fig 1 (what is the darker blue shading?), Fig 2 (the units of length on the X-axis are missing); Fig 5 (what types of error bars are shown?);
(6) lines 83-105: the descriptions of the ovarian stages would be better represented as a table rather than text in a paragraph.

Experimental design

I have 3 concerns about the experimental design that cast doubt on the validity of some of the study’s findings and conclusions:
(1) The study used fish that had been caught as by-catch in the fishery for another species of tuna in only a single season. As a result, only a small number of fish were studied, with histological evidence coming from 46 fish in one year. The sampling will not address the possibility of year-to-year variation in timing of spawning, and the sample size is substantially smaller than other recent studies of tuna spawning e.g. Ashida and Horie (2015) used 438 skipjack tuna; Farley et al (2015) used 640 southern bluefin tuna, and Okochi et al (2016) used 1040 Pacific bluefin tuna. The small sample sizes, and the limited spatial extent of the sampling do not provide sufficient information about the relative importance of spawning by this species in Mexican waters.
(2) The conclusion that the southern area of the Gulf of Mexico is ‘a new possible spawning area’ cannot be justified because sampling was not done extensively enough to show that this area is sufficiently distant from the spawning area in the northern Gulf of Mexico i.e. there could be a single large spawning area that covers the entire Gulf of Mexico. The data in the study only show that spawning appears to occur in Mexican waters.
(3) Sampling occurred only between January and April. Therefore the study’s conclusion #1 (line 160) that there is a ‘marked seasonal occurrence’ is not justified. The results show that the species was present for the months sampled but there is no data to show that the species was not present in other months.
Overall, the study provides some preliminary evidence that suggests Atlantic Bluefin tuna spawn in Mexican waters.

Validity of the findings

Covered in previous comments

Additional comments

This manuscript reports the results of a study into the reproductive status of Atlantic bluefin tuna in Mexican waters in the Gulf of Mexico. Fish were collected, as by-catch of the yellowfin tuna fishery, over 4 months in one year and their maturity and gonad state assessed histologically. While i can foresee the need and importance of this study, it was not suitably described in the text. The authors made a number of conclusions about from the limited data set on the timing of migration, the existence of a previously unreported spawning area, and the timing of spawning. The study was undertaken in a scientific manner; however the limited number of samples, and the spatial and temporal extents of the sampling, make this a preliminary study and do not sufficiently justify the authors’ conclusions.

Reviewer 2 ·

Basic reporting

in the Introduction section, it is missing some more details about why this area has been previously considered as potential spawning area for bluefin tuna (Abad-Uribarren et al. 2014), for example the occurrence of this species and suitable spawning window related to shift/evolution of oceanographic condition (More details in the attached document).

Experimental design

Overall, very few samples and few months were sampled (More details in the attached document).

Validity of the findings

The obtained results can show a posible new spawning area in southern Gulf of Mexico, as it is noted in the title of the MS. However, the few individuals analyzed it weak point of this study and thus requires to be cautious in the conclusions.

Additional comments

The MS #33905 – “A possible new spawning area for bluefin tuna (Thunnus thynnus): First histologic evidence of reproductive activity in southern Gulf of Mexico” provides information on bluefin tuna (Thunnus thynnus) ovary morphology in different reproductive stages in the southern Gulf of Mexico.
The investigation is primarily focused on microscopic analysis to assess the reproductive development of ovaries and testes. According to the knowledge of this reviewer, the classification methodology used for microscopic analysis is correct but certain concepts in the classification method should be clarified. But the main concern is the low number of samples analyzed and especially those in spawning that finally support the evidence that bluefin tuna conduct spawning activity in the area proposed.

Overall, the article is written in clear English, unambiguous, and it is a technically correct text. The article has conformed to professional standards of courtesy and expression, although certain clarification is required in the Material and Methods section. Overall, the article is concise and to the point and although the context of the study is well set and it includes sufficient literature references, there are missing some more details in the introduction and discussion section about why this area has been previously considered as potential spawning area for bluefin tuna (Abad-Uribarren et al. 2014), for example the occurrence of this species and suitable spawning window related to shift/evolution of oceanographic condition. In the opinion of the reviewer, this will better set the context/rationale of the study and the comparison with northern spawning areas, more than just the observations of different spawning areas in the Mediterranean Sea. The relevance and usefulness of the figure 2 and 5 it is not clear. In contrast figure 1, 3 and 4 and tables 1 and 2 are relevant for the content of the article.

More details about above mentioned are described below:

INTRODUCTION:
As mentioned previously, in the Introduction section, it is missing some more details about why this area has been previously considered as potential spawning area for bluefin tuna (Abad-Uribarren et al. 2014), for example the occurrence of this species and suitable spawning window related to shift/evolution of oceanographic condition. In the opinion of the reviewer, this will better set the context/rationale of the study and the comparison with northern spawning areas, more than just the observations of different spawning areas in the Mediterranean Sea.

MATERIAL AND METHODS
The classification method applied to define mature from immature individuals is not clear and further clarification is required. The authors stated that “females were classified as mature if ovaries contained vitellogenic, MG or HY oocytes and/or atresia”. However, it is not described which stage of vitellogenesis (early or advance) set the limit. As it is written, it seems that E-Vit is delimiting the threshold between immature/mature, and this is not what Schaefer (1998) and Farley et al. (2013) described in their studies. In both papers presence of A-Vit oocytes was the evidence of mature or active females (in Farley et al. 2013 and Schaefer 1998, respectively). This also contribute to the confusion when in line 84 it is defined as developing, ovaries contained E-Vit oocytes as MAGO.

According to the line 82-83 these are mature individuals. A table with histological classification criteria similar to Table 2 presented in Farley et al., 2013 is recommended. This table will also help authors defining the % of atresia used to determine the sexual maturity as it was done in Farley et al. (2013) and Schaefer (1998).

Lines 83-84 and 96-97 seem to be more result and should be moved to corresponding section.

RESULTS
In lines 114-117 I would suggest including number of individuals, as having only the % does not show the few number of individuals analyzed and this information is important for the interpretation of the results and how strength of the conclusions.

In line 114 change “a few…” to one individual. There is only one individual in developing according to the table 1. In line 117 also add how many (n=1) individuals were found at spawning.

Authors did not note if POF were present in analyzed ovaries. I would suggest adding information in this respect as if there were found could better support the hypothesis of a new spawning area in southern Gulf of Mexico.

Table 1 and 2. Could the authors provide the reason to divide the months in early, mid and late periods? How can this improve the analysis of the samples?

There is missing (probably in the Discussion section) the interpretation of the different maturity stages found in female and male BFT along the months analyzed and its relation with the reproductive cycle of this species. 45% of the individuals are in regenerating stage in the months previous to the individuals found spawning capable and spawning stages.

DISCUSSION
In the first paragraph of this section (lines 127-134) the authors support the arrival of the bluefin tuna to the study region by the timing of the catch. But also, due to increased feeding behavior before the spawning season, however, this study does not provide any evidence (e.g., stomach content data) to support this statement or literature references of previous studies where feeding behavior of bluefin tuna was described and thus support the hypothesis that bluefin tuna arrive to the area between January and March to increase feeding before the spawning season.

In paragraph two (lines 135-139) could be further develop the point described in Abad-Uribarren et al. 2014 where the suitable thermal and productive condition is related to potential bluefin tuna reproductive activity. And then compare with the conditions found in the northern Gulf of Mexico during the same period.

In paragraph tree (lines 140-155) and specially in lines (146-150) the authors consider as new spawning area the southern Gulf of Mexico based on the data analyzed. I would suggest to be more conservative in the statement considering there is only one female in spawning and that there was not found other spawning markers in other ovaries from January to April.

CONCLUSION:
Considering the few individuals analyzed and the only one female in spawning found in the study I would start the section with the last proposed sentence “Additional studies are needed …” and then “However… in the present study four facts provide evidence for considering…”. And again I would be very cautious in the fourth point as only one female was found in spawning and there were not found any other spawning markers such as POF that could better support the spawning activity of bluefin tuna in the southern Gulf of Mexico.

---

## Round 0.2 · accepted · Accept

Your edits have satisfied the points raised by reviewers.

Please incorporate the final corrections mentioned by Rev 2, and in the attached PDF, while in production

Reviewer 1 ·

Basic reporting

The authors have adequately addressed, in the revised manuscript, my comments relating to their reporting of the study.

Experimental design

The authors have adequately addressed my comments about the experimental design in their rebuttal and in the revised manuscript.

Validity of the findings

In response to my review of the original manuscript, the authors have adequately discussed the validity of their study, and its findings.

Additional comments

I am satisfied that the authofrs have adequately addressed my review comments, and their responses in the rebuttal are satisfactory.

Reviewer 2 ·

Basic reporting

Please check general comments

Experimental design

Please check general comments

Validity of the findings

Please check general comments

Additional comments

The MS #33905 – “A possible new spawning area for bluefin tuna (Thunnus thynnus): First histologic evidence of reproductive activity in southern Gulf of Mexico” provides information on bluefin tuna (Thunnus thynnus) ovary morphology in different reproductive phases in the southern Gulf of Mexico.

The investigation is primarily focused on microscopic analysis to assess the reproductive development of ovaries and testes. According to the knowledge of this reviewer, the classification methodology used for microscopic analysis is correct and the concerns raised during the first revision have been well addressed by authors (adding new tables). However, the main concern is still the low number of samples analyzed and especially those in spawning that finally support the evidence that bluefin tuna conduct spawning activity in the area studied. The authors, following reviewers suggestions, have been more cautious in their statements and further researcher has been requested in order to support the evidence found.

Overall, the article is written in clear English, unambiguous, and it is a technically correct text. The article has conformed to professional standards of courtesy and expression, and the authors have provided required clarifications in the Material and Methods section.

Overall, the article is concise and to the point and the context of the study is well set. The authors provided further explanations (literature references, e.g., Abad-Uribarren et al. 2014) in the introduction section about why this area has been previously considered as potential spawning area for bluefin tuna.

In my opinión, although the modification of Tables 1-3 can provide more information, there is still missing the interpretation of the occurrence of different ovary reproductive phases in certain months and its link with the reproductive cycle of the species. For example why 45% of the individuals are in regenerating phase just before the month when individuals were found spawning capable and actively spawning sub-phase.

Minor comments:
In line 102: change gonad by ovary
In line 105: change gonad by testes
I would suggest authors to keep the term reproductive "phase" when making reference to ovarian reproductive phases instead of using the term "stage", that would correspond more to oocyte development stages.